# Explicitly Maintaining Diverse Playing Styles in Self-Play

## Abstract

Self-play has proven to be an effective training schema to obtain a high-level agent in complex games through iteratively playing against an opponent from its historical versions. However, its training process may prevent it from generating a well-generalised policy since the trained agent rarely encounters diversely-behaving opponents along its own historical path. In this paper, we aim to improve the generalisation of the policy by maintaining a population of agents with diverse playing styles and high skill levels throughout the training process. Specifically, we propose a bi-objective optimisation model to simultaneously optimise the agents' skill level and playing style. A feature of this model is that we do not regard the skill level and playing style as two objectives to maximise directly since they are not equally important (i.e., agents with diverse playing styles but low skill levels are meaningless). Instead, we create a meta bi-objective model to enable high-level agents with diverse playing styles more likely to be incomparable (i.e. Pareto non-dominated), thereby playing against each other through the training process. We then present an evolutionary algorithm working with the proposed model. Experiments in a classic table tennis game *Pong* and a commercial role-playing game *Justice Online* show that our algorithm can learn a well generalised policy and at the same time is able to provide a set of high-level policies with various playing styles.

## 1 Introduction

Recent years have witnessed impressive results of self-play for Deep Reinforcement Learning (DRL) in sophisticated game environments such as various board (Silver et al., 2016; 2017a; Jiang et al., 2019) and video games (Jaderberg et al., 2019; Vinyals et al., 2019; Berner et al., 2019). The idea behind self-play is using a randomly initialised DRL agent to bootstrap itself to high-level intelligence by iteratively playing against an opponent from its historical versions (Silver et al., 2016). However, the training process of self-play may prevent it from obtaining a well-generalised policy since the trained agent rarely encounters diversely-behaving opponents along its own historical path. This can easily be taken advantage of by human players. Taking OpenAI Five (Berner et al., 2019) as an example, it has a win rate of 99.4% in more than 7000 Dota 2 open matches[1], but the replay shows that 8 of the top 9 teams that defeated the OpenAI Five are the same team and the policies they use in each game are very similar. This indicates that despite the remarkable high performance, the OpenAI Five agent is still not fully comfortable in some circumstances, which can be found and further exploited by human players (e.g., through meta-policies).

Generally, an agent can be identified from two aspects, skill levels and playing styles (Mouret & Clune, 2015). These two aspects are crucial for learning a high-level agent in the self-play training process because only playing against opponents with diverse playing styles and appropriate skill levels (i.e., not too low) can maximise the gains of the learning. If one only considers opponents' skill levels, there can be a catastrophic forgetting problem in which the agent "forgets" how to play against a wide variety of opponents (Hernandez et al., 2019). On the other hand, if one only considers playing styles, there will be a lot of meaningless games that the agent learns very little from its far inferior opponents (Laterre et al., 2018).

---

[1] https://arena.openai.com/#/results

Unfortunately, it is intrinsically challenging to strike a good balance between skill levels and playing styles in self-play. During the training process, the network weights of the DRL agent are usually optimised by a gradient-based method, which progresses along a single path that relies on the random seed of the network and the environment. At each iteration, the incumbent agent is the only response-policy for its historical versions. Such single-path optimisation is very unlikely to experience sufficiently diverse opponents, especially within a sophisticated environment. This may make common self-play algorithms, which use a probability function to decide which opponents to consider (e.g., the latest opponent or the past versions (Berner et al., 2019; Oh et al., 2019)), unable to generalise their policies, i.e., struggle to cope with opponents that are very different from which they have encountered before.

A viable way to introduce diverse playing styles in self-play is to consider the population-based approach, where a population of agents/opponents are maintained during the training process, with each potentially representing one play style. The population-based approach has already been frequently used in DRL (Jung et al., 2020; Carroll et al., 2019; Parker-Holder et al., 2020; Zhao et al., 2021). For example, Population-Based Training (PBT) (Jaderberg et al., 2017; 2019; Li et al., 2019; Liu et al., 2019) optimises a population of networks at the same time, allowing for the optimal hyperparameters and model to be quickly found. Neuroevolution (Heidrich-Meisner & Igel, 2009; Such et al., 2017; Salimans et al., 2017; Stanley et al., 2019) uses population-based evolutionary search (e.g., genetic algorithm and evolution strategy) to generate the agents' network parameters and topology.

In these population-based methods, an interesting idea to promote diversity of the agents' behaviours is to proactively search for "novel" behaviours. This can be very useful since maintaining a population of behaviours does not necessarily mean diversifying them over the search space (Jaderberg et al., 2019). This is particularly true in sparse/deceptive reward problems (Salimans et al., 2017; Conti et al., 2018) where the reward function may provide useless/misleading feedback leading to the agent to get stuck and fail to learn properly (Lehman & Stanley, 2011; Ecoffet et al., 2019). Such proactive-novelty-search techniques include novelty search (Conti et al., 2018; Lehman & Stanley, 2011), intrinsic motivation (Bellemare et al., 2016), count-based exploration (Ostrovski et al., 2017; Tang et al., 2017), variational information maximisation (Houthooft et al., 2016), curiosity-driven learning (Baranes & Oudeyer, 2013; Forestier et al., 2017), multi-behaviour search (Mouret & Doncieux, 2009; Shen et al., 2020) and quality-diversity (Cully & Demiris, 2018). They, based on the history information in the environment, motivate the agent to visit unexplored states in order to accumulate higher rewards (Conti et al., 2018; Ecoffet et al., 2019; Guo & Brunskill, 2019). For example, the quality-diversity algorithms use domain dependent behaviour characterisations to abstractly describe the agent's behaviour trajectory and encourage the agent to uncover as many diverse behaviour niches as possible, with each niche being represented by its highest-level agent (Mouret & Clune, 2015; Pugh et al., 2016). However, such proactive novelty search may not always be promising since novel behaviours that we search for do not always come with high skill levels. When it comes to population-based self-play, a game could be meaningless when the difference of agents' skill levels is too big, albeit their playing styles being very different. Indeed, what we need effectively is a population of high-level and diverse-style agents which play against each other through the training process.

To this end, this paper proposes a novel Bi-Objective (BiO) optimisation model to optimise skill levels and playing styles. One feature of this model is that we do not regard these two aspects as objectives to maximise directly, but rather we create a meta bi-objective model to enable high-level agents with diverse playing styles more likely incomparable (i.e. Pareto nondominated to each other), thus being always kept in the training process. Specifically, in BiO each objective is composed of two components. The first component is related to skill level of the agent, same for the two objectives, while the second component is related to playing style of the agent, we making it completely conflicting for the two objectives. As such, the Pareto optimal solutions in BiO are typically those far away from each other in playing styles but all with reasonably good skill levels (this will be explained in details in Section 3).

We propose an evolutionary algorithm to work with the proposed model. We follow the basic framework of multi-objective evolutionary algorithms, but with customized components for self-play.

We evaluate our algorithm in a classic table tennis game Pong and a commercial online role-playing game Justice Online [2].

It is worth mentioning that the problem here we are dealing with is different from multiobjective optimisation-related RL (Yang et al., 2019; Liu & Qian, 2021; Xue et al., 2022), such as those in the Multi-Objective Reinforcement Learning (MORL) (Moffaert & Nowé, 2014; Mossalam et al., 2016). MORL is a generalisation of standard reinforcement learning where the scalar reward signal is extended to multiple feedback signals, whereas our problem is to simultaneously optimise the skill levels and playing styles of the players.

## 2 PRELIMINARIES AND RELATED WORK

Each iteration of self-play can be considered as a Multi-agent Markov Decision Process defined by a tuple $\langle \mathcal{N}, \mathcal{S}, \{\mathcal{A}^i\}_{i \in \mathcal{N}}, \mathcal{P}, \{r^i\}_{i \in \mathcal{N}}, \gamma \rangle$, where $\mathcal{N} = \{1, \cdots, N\}$ is the set of $N > 1$ agents, $\mathcal{S}$ is the state space observed by all the agents, and $\mathcal{A}^i$ is the action space of agent $i$. Let $\mathcal{A} := \mathcal{A}^1 \times \cdots \times \mathcal{A}^N$, then $\mathcal{P} : \mathcal{S} \times \mathcal{A} \times \mathcal{S} \to [0, 1]$ is the transition probability from any state $s \in \mathcal{S}$ to any state $s' \in \mathcal{S}$ for any joint action $a \in \mathcal{A}$. $r^i : \mathcal{S} \times \mathcal{A} \times \mathcal{S} \to \mathbb{R}$ is the reward function that determines the immediate reward received by agent $i$ for a transition from $(s, a)$ to $s'$. $\gamma \in (0, 1]$ is the discount factor. At step $t$, each agent $i \in \mathcal{N}$ executes an action $a_t^i$, according to the system state $s_t$. The system then transitions to state $s_{t+1}$, and rewards each agent $i$ by $r^i(s_t, a_t, s_{t+1})$. The goal for agent $i$ is to maximise its own long-term reward $J_i(\pi) = \mathbb{E}_\pi \left[ \sum_{t=0} \gamma^t r^i(s_t, a_t, s_{t+1}) \right]$, by finding the policy $a_t^i \sim \pi^i(\cdot | s_t)$.

PROXIMAL POLICY OPTIMISATION (PPO). PPO (Schulman et al., 2017) is a popular deep policy gradient method where policy updates are computed by a surrogate objective regularised by the clipped probability ratios. Inspired by a trust region method, the algorithm updates the policy within a close neighbourhood around the previous-iteration policy each time to guarantee monotonic performance improvement. As shown in a clipped surrogate objective $L_{\pi_{\mathrm{old}}}^{\mathrm{CLIP}}(\pi)$:

$$L_{\pi_{\mathrm{old}}}^{\mathrm{CLIP}}(\pi) = \mathbb{E}\left[ \min \left( \frac{\pi(a \mid s)}{\pi_{\mathrm{old}}(a \mid s)} A^{\pi_{\mathrm{old}}}(s, a), \mathrm{clip}\left( \frac{\pi(a \mid s)}{\pi_{\mathrm{old}}(a \mid s)}, 1 - \varepsilon, 1 + \varepsilon \right) A^{\pi_{\mathrm{old}}}(s, a) \right) \right], \tag{1}$$

where $\mathrm{clip}(\cdot)$ removes the incentive of the probability ratio $\frac{\pi(a|s)}{\pi_{\mathrm{old}}(a|s)}$ outside the interval $[1 - \varepsilon, 1 + \varepsilon]$ ($\varepsilon = 0.2$). PPO can sample data from the stable previous-iteration policy $\pi_{\mathrm{old}}$, and incrementally refines the policy using multiple steps of stochastic gradient ascent before sampling new data.

MULTI-OBJECTIVE OPTIMIZATION. Multi-objective optimisation (Matthias, 2006) is an optimisation scenario that considers multiple objectives/criteria simultaneously. Without loss of generality, let us consider a maximisation scenario. Formally, a multi-objective optimisation problem can be expressed as:

$$\text{maximise } F(x) = (f_1(x), \cdots, f_m(x))^T, \tag{2}$$

where $m$ is the number of objectives. In the context of multi-objective optimisation, a solution $x_1$ is said better than $x_2$, if and only if $x_1$ is not worse than $x_2$ for all the objectives and better for at least one objective. We call this "better" relation as Pareto dominance, i.e., $x_1$ (Pareto) dominates $x_2$. Two solutions being incomparable means that they are non-dominated to each other. For a solution $x \in X$, if there is no solution in the solution set $X$ dominating $x$, then $x$ is called a Pareto optimal solution in $X$.

RELATED WORK. Diversifying playing styles of opponents that the agent encounters in the training process is an important topic in self-play. Interesting attempts include using diverse expert data as opponents to enrich the agent's experiences (Silver et al., 2016; Vinyals et al., 2019; Lowe et al., 2020), learning diverse sub-policies and combining them into an ensemble model (Xu et al., 2018), using reward shaping to create a series of different playing styles (Oh et al., 2019), and exploring unseen playing styles by domain randomisation (Jaderberg et al., 2019; Berner et al., 2019). However, striking a good balance between playing styles and skill levels is a challenging task; some of these methods need extra resources (e.g. expert data (Silver et al., 2017b)) or human hand engineering

---

[2]`https://mmos.com/review/justice`.

(e.g. manual tuning of the reward weights (Oh et al., 2019)), and some may degrade in sophisticated environments (Xu et al., 2018).

Another way to maintain diverse opponents in self-play is to consider the population-based training (PBT) (Jaderberg et al., 2019; Vinyals et al., 2019; Hernandez et al., 2019), where the playing style of each agent in the population is controlled by its own hyperparameters and reward weights. In such methods, PBT is to meta-optimise the internal rewards and hyperparameters of the agents. Since each agent in the population learns from the experience generated by playing against opponents sampled from the population, more generalised policies can be learned (Bansal et al., 2018; Zhao et al., 2021). However, maintaining an agent population does not necessarily mean maintaining diverse playing styles; thus it would be beneficial for this model to proactively explore novel playing styles and then properly maintain them during the training process. We will compare our model with PBT in the experimental studies.

It is necessary to note that multi-objective optimisation models, despite not designed for self-play, have been used to diversify agents' behaviours in games (Agapitos et al., 2008; Mouret & Doncieux, 2009; Shen et al., 2020; Pierrot et al., 2022b). They created one or several behaviour-related objectives for the algorithm to optimise directly. For example, the novelty of the agents' behaviour is considered as an auxiliary objective in Mouret & Doncieux (2009), and opposite behaviours (i.e., aggressive and defensive) of the agents are considered as two objectives in Shen et al. (2020). One main issue of such models is that an agent with a different behaviour from the rest will always be considered Pareto optimal no matter how poor its skill level is (since no other agent performs better in the corresponding behaviour objective). This may easily lead to the population to have diverse playing-style agents, but their skill levels can be highly variable. We will compare our model with one recent representative of such models in the experimental studies.

Lastly, it is worth mentioning that some recent studies formalised skills and styles with differentiable functions and leveraged gradient information to efficiently improve the skill level of agents from different stylistic directions (Fontaine & Nikolaidis, 2021; Pierrot et al., 2022a; Tjanaka et al., 2022), although they are not implemented in the self-play paradigm.

## 3 PROPOSED MODEL

Our bi-objective (BiO) optimisation model is designed to optimise skill levels of the agents and at the same time to diversify their playing styles. To do so, each objective is constructed to consist of two components. The first is the agent's skill level, which is shared by the two objectives. The second component is concerned with the agent's playing style, which we design to be opposite in the two objectives in order to make diverse style agents likely incomparable (thus as the Pareto optimal agents kept during the search). Formally, the BiO model for an agent $\pi$ with the skill level $g(\pi)$ and the playing style $h(\pi)$ is expressed as follows:

$$\text{maximise} \begin{cases} f_1(\pi) = g(\pi) + \kappa h(\pi) \\ f_2(\pi) = g(\pi) - \kappa h(\pi) \end{cases} \tag{3}$$

where $\kappa$ is a coefficient to make $g(\pi)$ and $h(\pi)$ commensurable. $g(\pi)$ is an indicator to reflect the agent's skill level, and it can be represented by win rate (Chen et al., 2018), Elo score (Jaderberg et al., 2019), etc. $h(\pi)$ is an indicator to reflect the agent's playing style, and its calculation will be explained later. The difference between two agents in $h(\pi)$ indicates how differently they behave, e.g., aggressively versus defensively.

Figure 1 gives an example to illustrate the proposed model, where Figure 1(a) presents six agents in the original skill-style space and Figure 1(b) presents them in the proposed bi-objective space. To help understand the characteristics of the BiO, we provide the following remarks, which can be derived from the figure as well as Equation 3.

- A higher skill-level agent will not be dominated by a lower level one in the proposed model. This can be derived from Equation 3 — if $g$ is higher for the agent $\pi_1$ than $\pi_2$, then whatever their playing styles $h(\pi_1)$ and $h(\pi_2)$ are, $\pi_2$ will not be better than $\pi_1$ on both objectives $f_1$ and $f_2$; in the best case for $\pi_2$, they are non-dominated to each other (e.g., the agents **A** and **B** in Figure 1).
- Similar playing style agents tend to be comparable (i.e., dominating or being dominated) even if their skill levels are close. For example, in Figure 1 the agent **E**'s playing style

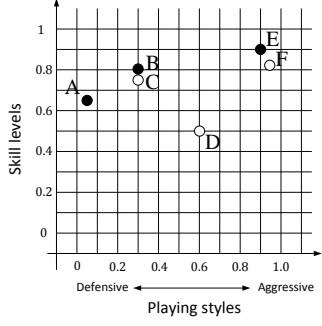
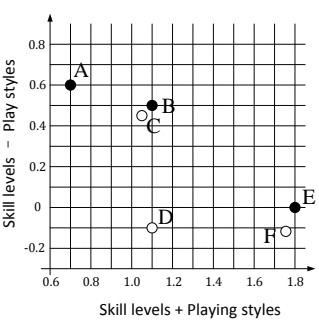

(a) The original skill-style space          (b) The proposed BiO space

Figure 1: An illustration that the proposed bi-objective model makes diverse style agents with fairly good skill level more likely to be Pareto optimal (i.e., **A**, **B**, **E**). As implemented in this paper, the skill level and playing style are normalized and the coefficient $\kappa = 1$. (a) The six agents in the skill-style space. (b) The six agents in the proposed BiO space where the two meta-objectives are to be maximised. As can be seen in Figure 1(a), the agents **B** and **C** have the same playing style, while **B** has a higher skill level than **C**. The agents **E** and **F** have similar styles, while **E** has a higher skill level than **F**. **A**'s style is dissimilar from the others' and the same for **D**, while **A**'s skill level is significantly better than **D**'s. Therefore, if one would like to choose three diverse style agents with good skill levels as an opponent population for an agent to play against, the agents **A**, **B** and **E** can be the best choices. This, as shown in Figure 1(b), is in line with the proposed model, where these three agents are Pareto optimal in the bi-objective space.

is similar to **F**'s and their skill levels are fairly close, but **E** dominates **F** in the proposed model. In the special case that two agents have the same style, the higher skill level agent always dominates the lower one (e.g. the agents **B** and **C** in the figure).

- Dissimilar playing style agents tend to be incomparable (i.e., non-dominated to each other) even if one is fairly inferior to the other in skill level. For example, the agent **A** in Figure 1, which has relatively low skill level, is not dominated by any agent since its style is rather different from the others'. In fact, it can be derived from Equation 3 that the probability of two agents being incomparable increases linearly with the distance of their playing styles.

- The coefficient $\kappa$ in the model is a critical parameter which weighs up between agents' skill level and playing style. In our implementation, we simply set $\kappa = 1$ after normalising them (i.e. after making $g(\pi) \in [0, 1], h(\pi) \in [0, 1]$). It is worth mentioning that a different $\kappa$ setting may potentially be more suitable for a specific problem.

To sum up, the proposed BiO model enables different style agents with good skill level to be ranked high. As can be seen in Figure 1, if one would like to choose three out of the six agents as the population of opponents for an agent to play against, then the agents **A**, **B** and **E** will be chosen (since they are Pareto optimal). **B** and **E** are chosen because they have higher skill levels than their similar agents (i.e., **C** and **F**, respectively). **A** is chosen because **A** is far away from the others in playing style, whereas **D** is not chosen because despite being far away from the others, but its skill level is significantly worse than them too. In fact, the model can be seen to identify representative good agents, but without a need of setting a niche. As such, one can maintain a population of high-level agents with diverse playing styles who keep playing against each other throughout the training process.

In BiO, the playing style $h(\pi)$ of an agent $\pi$ can be estimated by $\pi$ playing against all the opponents in the pool. Formally, it can be calculated as

$$h(\pi) = \frac{1}{K} \sum_{k=1}^{K} \frac{1}{T_k - 1} \sum_{t=1}^{T_k - 1} var_{\pi_k}(s_t, s_{t+1}) \tag{4}$$

where $K$ is the size of the opponent pool, $T_k$ is the length of the trajectory produced by $\pi$ playing against the opponent $\pi_k$, and $var_{\pi_k}(s_t, s_{t+1})$ denotes the relevant state change (e.g. the position change of the agent $\pi$) between time steps $t$ and $t + 1$.

Based on the gameplay of games, the calculation of the state change function can generally be classified into two categories, opponent-independent mode and opponent-dependent mode. In the

opponent-independent mode, the function can directly be estimated by the change of the agent $\pi$'s own states. This mode is for games where the players have their own venues, for example the games Pong, Tennis and Blobby Volley. In such games, the movement frequency of the agent $\pi$ is the main way to reflect its playing style — aggressive players frequently change their positions whereas conservative players tend to move slowly and steadily.

In the opponent-dependent mode, the state change function is estimated by not only the agent $\pi$'s states but the opponent $\pi_k$'s. That is, the opponent's states are used as a reference to estimate the agent's playing style. This mode is for games where the agent and its opponent share the same venue, for example the battle games Justice Online, B&S Arena Battle (Oh et al., 2019) and Toribash (Kanervisto & Hautamäki, 2019). In such games, the agent's playing style needs to be measured by the difference between the agent's movement trajectory and its opponent's movement trajectory — aggressive players like close combat, thus the difference typically small, whereas defensive players prefer to move around and tend to stay in a certain distance from their opponents, thus the difference typically large.

## 4 OPTIMISATION

In this section, we present an evolutionary algorithm working with the Proximal Policy Optimisation (PPO) (Schulman et al., 2017) to optimise the proposed bi-objective model. Specifically, we use the framework of the classic multi-objective evolutionary algorithm NSGA-II (Deb et al., 2002), but with customized components for self-play DRL, e.g., an evaluation population "frozen" for a while allowing the agents to play against.

Algorithm 1 gives the main procedure. The procedure starts by initialising the evolutionary population with a set of agents defined by random neural networks ($\theta$) with reward weights ($\omega$). Other hyperparameters of the agents (e.g. neural architecture, discount factor and learning rate) are set manually and can be found in Appendix C. Step 1 in the algorithm is to estimate the skill level and playing style of the evolutionary population. Each of its agents plays against all the agents of the evaluation population and obtains the average skill level and playing style. Then the agents' fitness (i.e. objective functions) of the proposed BiO model is calculated in Step 2. After that, the following steps repeat for each generation of the evolutionary algorithm. Step 4 is to update the population by varying the agents' reward weights. Here, we adopt two basic real-valued variation operators, simulated binary crossover and polynomial mutation (Deb et al., 2002) (details can be seen in Appendix B). After new agents generated, we use PPO to train their network parameters (Steps 5 and 6), where each new agent randomly picks one agent in the

**Algorithm 1:** Algorithm to solve the bi-objective model

---

**Input:** Evolutionary population
$P \leftarrow \langle\theta_1,\omega_1\rangle, ..., \langle\theta_K,\omega_K\rangle$; Evaluation population $E \leftarrow P$; Generation $gen \leftarrow 0$

1   $P \leftarrow Evaluation(P, E)$;
2   $P \leftarrow FitnessAssignment(P)$;
3   **while** *termination condition not met* **do**
4     $P' \leftarrow Variation(P)$;
5     **foreach** $\langle\theta',\omega'\rangle \in P'$ **do**
6       $\theta' \leftarrow PPO(\theta',\omega',P)$;
7     $P' \leftarrow Evaluation(P', E)$;
8     $P' \leftarrow FitnessAssignment(P')$;
9     $P \leftarrow EnvironmentalSelection(P, P')$;
10    $gen \leftarrow gen + 1$;
11    **if** $gen \% freq = 0$ **then**
12      $E \leftarrow P$;
13      $P \leftarrow Evaluation(P, E)$;
14      $P \leftarrow FitnessAssignment(P)$;

15   **return** $\langle\theta,\omega\rangle$ which has the highest skill level in $P$

---

parental population to play against and this repeats a few times. Then, the new agent population are evaluated against the evaluation population (Step 7) and their objective functions of the BiO model are calculated (Step 8). Next, based on the objective functions, the environmental selection is performed by using the nondominated sorting and crowding distance in NSGA-II (Deb et al., 2002) (details are given in Appendix B) to select the $K$ best solutions from the parental and new populations as the next-generation population (Step 9).

Steps 11 to 14 are to update the evaluation population after a while (i.e. every $freq$ generations). Introducing an evaluation population that lets the current population play against for estimating the skill level and playing style plays an important role. Fixing the opponents of the agents from different generations enables their fitness comparable; playing against different opponents can easily produce different skill level and playing style even for the same agent. Note that the update frequency

parameter $freq$ cannot be set very large/small. A large value may make the skill level of the evaluation population far lower than the current agents', while a very small value may make the fitness of the agents unstable during the evolutionary process.

Finally, after the procedure terminates, the returned agent is the one that has the highest skill in the evolutionary population (Step 15).

## 5 EXPERIMENTS

We evaluate our model on the Atari game Pong and the commercial game Justice Online. To implement the bi-objective model, the skill level in Pong is estimated by the difference of the final score between the agent and its opponent, and the playing style is done by the move of the agent's paddle. In Justice Online, the skill level is estimated by the difference in the final *health point* between the agent and its opponent, and the playing style by their average distance in the game. Details about the two environments and their BiO implementations are given in Appendix A.

We consider four competitors to compare with our proposed model; they are PPO (Schulman et al., 2017), a DRL strategy based on Population-Based Training (Jaderberg et al., 2017) (denoted by PBT for simplicity), a DRL strategy considering multiple rewards (Oh et al., 2019) (denoted by multi-reward), and a DRL strategy based on evolutionary multi-objective optimisation (called EMOGI) (Shen et al., 2020). Note that all of these competitors except multi-reward were not designed specifically for self-play; here you used their self-play versions, where the agent is trained by playing against an opponent from its historical versions. PPO is a baseline DRL method, upon which the other algorithms (including ours) are based. To alleviate the catastrophic forgetting problem, in PPO we follow the practice in (Berner et al., 2019) to let the agent to play against the latest version with a probability of $80\%$ and against past versions with a probability of $20\%$. The multi-reward competitor is PPO working with a multi-reward strategy (Oh et al., 2019) where multiple rewards are used in training to shape different playing styles (aggressive, balanced and defensive) for generalisation. The PBT competitor is PPO working under the population-based learning environment (Jaderberg et al., 2017), where each PPO style agent of the population can exploit information from the rest of the population. Like the proposed BiO, EMOGI (Shen et al., 2020) combines PPO with the evolutionary multi-objective optimisation approach, but the objectives are defined based on a hierarchical comparison of win rate and two handcrafted playing styles (i.e., aggressive and defensive). More details about these four competitors can be found in Appendix C.

For a fair comparison, all the four algorithms are run within the same computational budgets (1G frames) and the average win rate of 500 games for each environment is reported. In our algorithm, the frequency of updating the evaluation population $freq$ is set to 5. The hyperparameters of PPO for all the algorithms are set according to (Schulman et al., 2017). Appendix C gives all the parameters and their settings in the experiments.

We aim to evaluate the proposed algorithm through answering the following four questions.

- *Research Question 1 (RQ1)*: How does BiO perform, in comparison with the other algorithms, in the same training and evaluation environments when playing against a built-in AI?
- *Research Question 2 (RQ2)*: How does BiO perform, in comparison with the other algorithms, in different training and evaluation environments when against a built-in AI?
- *Research Question 3 (RQ3)*: What happens when the agents obtained by BiO and the other algorithms play directly against each other?
- *Research Question 4 (RQ4)*: What kind of agents can BiO provide? Do they have diverse playing styles?

The experiment for the question RQ1 is to see the ability of the obtained policy in playing against a rule-based built-in AI player when the training and evaluation environments are the same. The experiment for RQ2 is to see the generalisation of the policy to a different environment. Here, we change the start position of the opponent from the central point of the site in the training to a random boundary point in the evaluation. The above two experiments are all about playing against a build-in AI; then one would be interested to know what if the agents obtained by all the algorithms play directly against each other. RQ3 is designed to answer this question. The experiment for RQ4 is to

see if the proposed BiO model can produce diverse playing styles. Since Algorithm 1 returns only the highest skill-level policy, one may be curious about what the whole population (i.e., $P$ in Step 15 of the algorithm) looks like. RQ4 is for this question.

**RQ1: How does BiO perform in the same environment when against the built-in AI?** Table 1 gives the win rate of the policies obtained by the five algorithms. As can be seen in the table, PBT has the best overall performance. This may be attributed to the fact that PBT configures optimal hyperparameters for the training to avoid premature convergence. Our algorithm BiO performs best in Pong and takes the second place in Justice Online, achieving a win rate of over $90\%$ for both games. In contrast, the agents trained by the other three algorithms perform differently in the two environments, especially for Multi-reward which only has close to $20\%$ win rate in Justice Online, but more than $97\%$ win rate in Pong. One explanation is that the built-in AI in Justice Online is rather different from the opponents that the three algorithms' agent (PPO, Multi-reward and EMOGI) encounters in the training, but in Pong they are very similar.

Table 1: Win rate of the policies against the built-in AI in the same training and evaluation environments.

|  | PPO | PBT | Multi-reward | EMOGI | BiO (ours) |
|---|---|---|---|---|---|
| Pong | 92.8% | 99.4% | 97.4% | 99.4% | 99.7% |
| Justice Online | 74.0% | 99.2% | 20.4% | 80.8% | 93.0% |

**RQ2: How does BiO perform in different environments when against the built-in AI?** Table 2 gives the win rate of the policies obtained by the five algorithms. It can be seen that the environmental changes have a greater impact on all the algorithms except BiO. In Pong, the performance of the four peer algorithms (PPO, PBT, Multi-reward and EMOGT) have a significant drop, especially the win rate of PPO, PBT and EMOGT being less than $50\%$. In Justice Online, these four algorithms still cannot obtain a good winning rate, despite a slight increase of the Multi-reward's performance compared with the result in the same training and evaluation environments (Table 1). In contrast, our BiO maintains a win rate above $80\%$ when the environment changes, even reaching over $90\%$ win rate in Justice Online. One explanation for this is that the built-in AI is affected very differently by changes in different environments.

Table 2: Win rate of the policies against the built-in AI in different training and evaluation environments.

|  | PPO | PBT | Multi-reward | EMOGI | BiO (ours) |
|---|---|---|---|---|---|
| Pong | 39.8% | 19.8% | 64.5% | 8.4% | 84.1% |
| Justice Online | 2.4% | 35.6% | 39.0% | 75.0% | 91.0% |

**RQ3: What happens when the agents obtained by BiO and the other algorithms play directly against each other?** In this experiment, we construct an opponent pool of the agents ever generated, i.e., one agent from PPO, one from PPT, three from multi-reward (as it generates three agents with aggressive, balanced and defensive playing styles respectively), 30 from EMOGI and 30 from BiO (the population size is 30). We then randomly pick one from the opponent pool to be played against our agent. This repeats 60 times and the average win rate is reported in Table 3. As shown, BiO achieves a significantly higher win rate than the other four algorithms (except in Justice Online with the same environment where PBT's win rate is slightly higher than BiO's), indicating better generalisation of the proposed model to different opponents. It is worth noting that the advantage of BiO over its competitors is more evident when training and testing in different environments. This implies the importance of maintaining a set of high skill-level agents with diverse styles during the training, which provides the agent with opportunities to play against very different opponents.

**RQ4: Can BiO provide diverse playing styles?** Since our algorithm (Algorithm 1 in Section 4) returns only the highest skill-level agent in the final population, one may ask how the remaining agents look like. Figure 2(b) plots the final population obtained by the algorithm under the Justice Online environment in the space of skill level and playing style. As can be seen, these agents have similar skill level and diverse playing styles. This indicates that our algorithm can provide agents with very different behaviours. These agents should all have good generalisation as they encountered the same set of various opponents in the training.

Table 3: Win rate of the policies against a pool of the opponents obtained by all the algorithms in the same and different training-evaluation environments.

|  | PPO | PBT | Multi-reward | EMOGI | BiO (ours) |
|---|---|---|---|---|---|
| Pong (same environment) | 54.6% | 59.0% | 52.4% | 26.9% | 83.5% |
| Justice Online (same environment) | 71.0% | 79.6% | 6.0% | 55.0% | 78.0% |
| Pong (diff. environment) | 60.7% | 39.3% | 70.9% | 42.4% | 88.8% |
| Justice Online (diff. environment) | 45.9% | 45.9% | 45.7% | 63.7% | 81.8% |

To get a sense of how the results look like in the game environments, Figures 2(c)-(e) plot the behaviour trajectories of three representative agents (aggressive, neural and defensive) tagged in Figure 2(b) on nine randomly selected games. As shown, although each agent has very similar behavioural trajectories (orange line) across the nine games, different agents behave rather differently. The aggressive agent runs towards the opponent, the defensive agent tends to avoid confrontation with the opponent and runs around the arena, and the neutral agent stays or walks around some of the arena's boundaries.

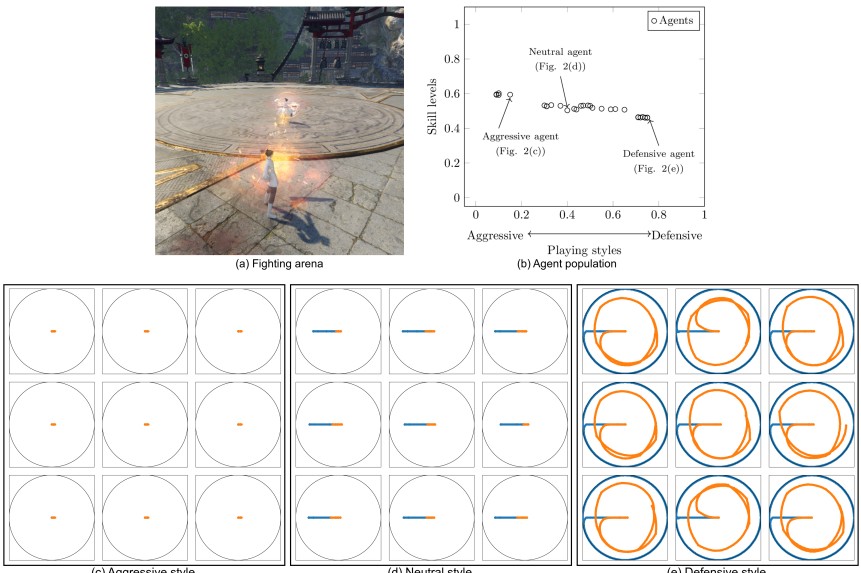

Figure 2: An illustration that BiO can obtain a set of agents with diverse play styles under the Justice Online environment. (a) A scene of the game environment. (b) The final agent population obtained by BiO in the space of skill level and playing style. (c)-(e) The behaviour trajectories of three representative agents tagged in Figure 2(b) on nine randomly selected games, where the orange and blue lines represent the footprints of the agent and opponent, respectively.

## 6 CONCLUSION

This paper proposes a bi-objective optimisation model (BiO), working with an evolutionary algorithm, to improve the generalisation of the policy in self-play. BiO maintains an agent population and enables high-level, diverse-style agents more likely to be Pareto optimal, thus playing against each other throughout the training process. The experimental studies have shown its effectiveness and robustness in different environments. In addition to the improvement of the policy's generalisation, a by-product from this model is that it can provide a set of high skill-level policies with diverse behaviours.

This work is the first step towards a new attempt of balancing performance and behaviour in games, and it could be potentially improved by different ways, for example, tuning the coefficient $\kappa$ of the model for specific problems and developing other population-based algorithms for the model. In addition, despite being used for self-play here, it is extendable to other RL settings where the agent's behaviour can be properly measured.

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

## A ENVIRONMENTS

This section describes the two environments *Pong* and *Justice Online* and the implementation of the bi-objective model BiO in them, i.e., how to determine the skill level function $g(\pi)$ and the playing style function $h(\pi)$ in the two environments.

PONG. Pong is a two-player game that simulates table tennis, in which each player controls an in-game paddle by moving it vertically across the left or right side of the screen. There are three actions that the players can take: move up the paddle, move down the paddle, or stay still. Points are earned when one fails to return the ball to the other. The goal of the game is for each player to reach 20 points before the opponent. The self-play version of Pong we used in the experiments is accessible at `https://github.com/xinghai-sun/deep-rl/blob/master/docs/selfplay_pong.md`.

To estimate the skill level of an agent in Pong, we consider the average difference of the final scores between the agent and its opponent in the opponent pool. Formally, the skill level function $g(\pi)$ of

the agent $\pi$ is calculated as:

$$g(\pi) = \frac{1}{K} \sum_{k=1}^{K} \frac{score_\pi - score_{\pi_k} + 20}{40} \tag{5}$$

where $K$ is the size of the opponent pool (i.e. the population size in our algorithm) and $score_\pi$ is the score of the agent $\pi$ at the end of the game.

The playing style in Pong is in the opponent-independent mode. It can be reflected by the move frequency of the agent's paddle. As such, the state change function $var_{\pi_k}$ (when $\pi$ plays against $\pi_k$) in the playing style function (i.e. Equation (3) in the text)

$$h(\pi) = \frac{1}{K} \sum_{k=1}^{K} \frac{1}{T_k - 1} \sum_{t=1}^{T_k - 1} var_{\pi_k}(s_t, s_{t+1}) \tag{6}$$

can be defined as

$$var_{\pi_k}(s_t, s_{t+1}) = \begin{cases} 0, & \text{if the position of } \pi\text{'s paddle in state } s_{t+1} \text{ is the same as in } s_t \\ 1, & \text{otherwise} \end{cases} \tag{7}$$

where $T_k$ denotes the length of the trajectory produced by $\pi$ playing against the opponent $\pi_k$.

JUSTICE ONLINE. Justice Online [3] supplies duels between two players called "Arena Battles". The arena battle is a two-player zero-sum game, where two players fight against each other to decrease the opponent's Health Point (HP) to zero as the goal within a fixed amount of time. This game is a non-trivial task. First, it is a real-time game which requires two players to make decisions simultaneously under imperfect information. At any time of the game, neither player knows what the opponent's current skills and skill levels are likely to be. Second, the game has a massive state space. An agent must make various skills, moves and targeted decisions simultaneously, leading to a huge number of possible states.

To estimate the skill level of an agent, we consider the average difference of the final HP values between the agent and its opponent in the opponent pool, expressed as:

$$g(\pi) = \frac{1}{K} \sum_{k=1}^{K} HP_\pi - HP_{\pi_k} \tag{8}$$

where $K$ is the size of the opponent pool and $HP_\pi$ is the HP value of the agent $\pi$ at the end of the game.

The playing style in Justice Online is in the opponent-dependent mode. It needs to be measured by the relative position (i.e. distance) between the agent and its opponent. Aggressive agents like close combat and their distance to the opponent is small, whereas defensive agents like to stay in a certain distance from their opponents. We thus define the $var_{\pi_k}$ function in Equation 6 as

$$var_{\pi_k}(s_t, s_{t+1}) = dist_{t+1}(\pi, \pi_k) \tag{9}$$

where $dist_{t+1}(\pi, \pi_k)$ denotes the Euclidean distance between the agents $\pi$ and $\pi_k$ at step $t + 1$. Note that both the skill level and playing style values are normalized into the range of $[0, 1]$ in our implementation.

## B   ALGORITHM DETAILS

This section first describes how to generate new agents in our algorithm and then explains how to constitute the new population based on the old population and newly-generated agents.

### B.1   AGENT GENERATION

In the proposed algorithm, variable parameters of the agents are the network weight $\theta$ and the reward weight $\omega$; other hyperparameters (e.g. neural architecture, discount factor and learning rate) are fixed and set manually (their settings can be found in Appendix C). Here, we consider basic variation operators from evolutionary algorithm for searching for promising $\theta$ and $\omega$. For $\theta$, we use the single-point crossover (Pawelczyk et al., 2018) to recombine two networks. It randomly selects a hidden layer of two networks and swaps part of them to produced two new networks. Figure 3 illustrates this operator.

---

[3] https://github.com/NeteaseFuxiRL/nsh

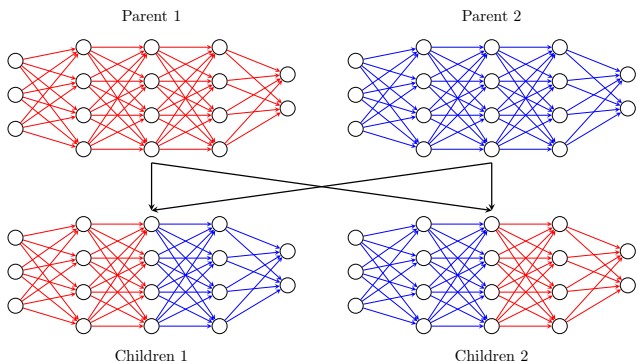

Figure 3: An illustration of the single-point crossover for generating new neural networks.

## B.2 ENVIRONMENTAL SELECTION

We use the classic NSGA-II environmental selection procedure (Deb et al., 2002) to select the $K$ best solutions as the next-generation population from the union of the old population $P$ and the new generated solutions $P'$, as shown in Algorithm 2. First, we divide the union into different fronts $(F_1, F_2, ..., F_i, ...)$ where the solutions in the same front are nondominated to each other. Then, the critical front $F_i$ is found such that $|F_1 \cup F_2 \cup ... \cup F_{i-1}| \leq K$ and $|F_1 \cup F_2 \cup ... \cup F_{i-1} \cup F_i| > K$; correspondingly the first $i - 1$ fronts $(F_1, F_2, ..., F_{i-1})$ make up the new population $P$. Now if the size of $P$ is less than the population capacity $K$, we calculate the crowding distance of the solutions in $F_i$ and sort them in a descending order. Next, we choose the first $K - |P|$ solutions of $F_i$ and put them into $P$. Now, $P$ is the population to be returned.

---

**Algorithm 2:** $EnvironmentalSelection(P, P')$ (Deb et al., 2002)

---

**Input:** Population capacity $K$;
1 $F_1, F_2, ..., F_i, ... \leftarrow NondominatedSort(P \cup P')$ ;   // Partition $P$ into different nondominated fronts $F_1, F_2, ..., F_i, ...$ and find the critical front $F_i$ where $0 \leq K - |F_1 \cup F_2 \cup ... \cup F_{i-1}| < F_i$
2 $P \leftarrow F_1 \cup F_2 \cup ... \cup F_{i-1}$ ;
3 **if** $|P| < K$ **then**
4 $\quad F_i \leftarrow CrowdingDistance(F_i)$ ;   // Calculate crowding distance of each solution in $F_i$
5 $\quad F_i \leftarrow Sort(F_i)$ ;   // Sort in the desending order accroding to crowding distance
6 $\quad P \leftarrow P \cup F_i[1 : K - |P|]$ ;// Choose the first $K - |P|$ solutions of $F_i$
7 **return** $P$

---

## C EXPERIMENT DETAILS

This section details all parameters/hyperparameters used in the experiments. We first list the PPO-associated hyperparameters (Mnih et al., 2016) which are shared by all the four algorithms. This is followed by the parameters/hyperparameters specific to each algorithm. Finally we give the reward settings of the four algorithms for the two environments Pong and Justice Online.

PARAMETERS FOR PPO USED IN ALL THE ALGORITHMS.

- **Discount Rate ($\gamma = 0.99$).** This parameter controls how much the RL agent cares about rewards in the immediate future relative to those in the distant future.

- **Weights for Loss Function** ($[1, 0.5, 0.01]$)**.** The loss function of PPO consists of three terms, the policy loss (actor), the value loss (critic), and an entropy loss. Their weights were assigned 1, 0.5 and 0.01, respectively.

- **Learning Rate for Adam** (0.00025). This parameter controls the learning rate of the loss function.
- **Neural Architecture** [256, 128, 128]. The network consists of three hidden layers, of which the first one has 256 nodes and the other two have 128 nodes. For the image input (such as Pong game), we stack 4 layers of $3 \times 3$ convolutions before the fully connected layers.

PARAMETERS FOR PPO-SP.

- **Opponent Pool Update Frequency** (1, 000 **episodes**). This parameter controls the update frequency of the opponent pool in self-play, i.e., how often the algorithm sends its agent into the opponent pool.
- **Opponent Change Frequency** (3 **episodes**). This parameter controls how long the agent plays against one opponent.

PARAMETERS FOR THE MULTI-REWARD ALGORITHM.

- **Opponent Pool Update Frequency** (1, 000 **episodes**). This parameter controls the update frequency of the opponent pool in self-play.
- **Opponent Change Frequency** (3 **episodes**). This parameter controls how long the agent plays against one opponent.
- **Opponent Selection Probability.** In the multi-reward algorithm, the most recent 10 models of each playing style are selected to play against the agent with a probability $p$, while the other past versions are selected uniformly with probability $1 - p$. As practiced in (Oh et al., 2019), $p$ was set gradually decreased from 0.8 to 0.1 with the progress of the training.

PARAMETERS FOR THE PROPOSED ALGORITHM.

- **Population Size** ($K = 30$). This parameter controls the size the population in the evolutionary process.
- **Single-Point Crossover Probability** (0.5). This parameter controls the probability of performing the recombination of two networks.
- **Simulated Binary Crossover Probability** (1.0). This parameter controls the probability of performing the recombination of two agents' reward weights. The distribution index $\eta_c = 20$ was used (Deb et al., 2002).
- **Polynomial Mutation Probability** ($p_m = 0.05$). This parameter controls the probability of performing the disturbance of an agent's reward weights. The distribution index $\eta_m = 20$ was used (Deb et al., 2002).
- **PPO Training Budget** ($Pong = 600, 000, Justice\ Online = 1, 000, 000$). This parameter controls the training overheads (frames) of PPO in one generation of the evolutionary algorithm.

REWARD SETTINGS IN PONG. In Pong, the internal reward at step $t$ for the algorithms PPO and PPO-SP was set to $r_t = 1$ if the agent wins at that step; otherwise $r_t = 0$. As for the algorithms multi-reward and BiO, different playing styles of agents are considered explicitly or implicitly. So a factor to reflect the agent's playing style was also included in the reward function. Formally, the internal reward $r$ at step $t$ is expressed as

$$r_t = w_1 r_t^{skill} + w_2 r_t^{style} \tag{10}$$

where

$$r_t^{skill} = \begin{cases} 1, & \text{if the agent wins at step } t \\ 0, & \text{otherwise} \end{cases} \tag{11}$$

and

$$r_t^{style} = \begin{cases} 1, & \text{if the position of the agent's paddle in state } s_t \text{ is different from that in } s_{t-1} \\ 0, & \text{otherwise} \end{cases} \tag{12}$$

$w_1$ and $w_2$ are two weight parameters. Among them, $w_2$ is to control agents' playing styles. In the multi-reward algorithm, $w_1$ was set to 1 across the three styles, while $w_2$ was set to 0.01, 0 and $-0.01$ for the busy, neutral and lazy styles, respectively. In our algorithm, $w_1$ and $w_2$ are generated by the evolutionary search (i.e. simulated binary crossover and polynomial mutation) within the range of $w_1 \in [0, 1]$ and $w_2 \in [-0.01, 0.01]$.

REWARD SETTINGS IN JUSTICE ONLINE.    In Justice Online, the goal of the agent is to decrease the opponent's Health Point (HP) while trying to keep its own HP non-decreasing. Therefore, the HP margin can be used as the internal reward. Specifically, in PPO and PPO-SP we considered the following internal reward at step $t$ when the agent $\pi$ plays against its opponent $\pi_k$.

$$r_t = (HP_t^\pi - HP_{t-1}^\pi) + (HP_{t-1}^{\pi_k} - HP_t^{\pi_k}) \tag{13}$$

where $HP_t^\pi$ denotes the HP value of the agent $\pi$ at step $t$. The term $(HP_t^\pi - HP_{t-1}^\pi)$ is to encourage the agent to defend the opponent's attack, and the term $(HP_{t-1}^{\pi_k} - HP_t^{\pi_k})$ is to encourage the agent to attack the opponent.

As for the algorithms multi-reward and BiO, since different playing styles of agents are considered, we added a term to reflect the agent's playing style. Specifically, we used the following internal reward at step $t$ when the agent $\pi$ plays against its opponent $\pi_k$.

$$r_t = w_1(HP_t^\pi - HP_{t-1}^\pi) + w_2(HP_{t-1}^{\pi_k} - HP_t^{\pi_k}) + w_3(dist_t(\pi, \pi_k) - dist_{t-1}(\pi, \pi_k)) \tag{14}$$

where $dist_t(\pi, \pi_k)$ denotes the Euclidean distance between the agents $\pi$ and $\pi_k$ at step $t$. Among the three weight parameters, $w_1$ and $w_2$ are concerned with the skill level while $w_3$ controls the playing style. In the multi-reward algorithm, $w_1$ and $w_2$ were set to 1 across the three styles, while $w_3$ was set to 1, 0 and $-1$ for the aggressive, neutral and defensive styles, respectively. In our algorithm, all the weights are generated by the evolutionary search within the range of $w_1 \in [0, 1]$, $w_2 \in [0, 1]$ and $w_3 \in [-1, 1]$. Table 4 and Table 5 summarize the reward weight settings of the multi-reward algorithm and the proposed algorithm in the two environments, respectively.

Table 4: Reward weights of each style in the multi-reward algorithm.

| Environments | Playing styles | Reward weights |
|---|---|---|
| Pong | busy | $w_1 = 1, w_2 = 0.01$ |
| | neutral | $w_1 = 1, w_2 = 0$ |
| | lazy | $w_1 = 1, w_2 = -0.01$ |
| Justice Online | aggressive | $w_1 = 1, w_2 = 1, w_3 = 1$ |
| | neutral | $w_1 = 1, w_2 = 1, w_3 = 0$ |
| | defensive | $w_1 = 1, w_2 = 1, w_3 = -1$ |

Table 5: The range of reward weights for the search in the proposed algorithm.

| Environments | Weight range |
|---|---|
| Pong | $w_1 \in [0, 1], w_2 \in [-0.01, 0.01]$ |
| Justice Online | $w_1 \in [0, 1], w_2 \in [0, 1], w_3 \in [-1, 1]$ |

