# OpenReview forum: "Explicitly Maintaining Diverse Playing Styles in Self-Play"
_ICLR.cc/2023/Conference — Submitted to ICLR 2023_

### Official Review · Reviewer_FdvX · 2022-10-23

**Confidence:** 3
**Correctness:** 3
**Technical Novelty And Significance:** 2
**Empirical Novelty And Significance:** 2
**Recommendation:** 3

**Clarity, Quality, Novelty And Reproducibility:**

- Clarity: The text is readable and not hard to follow.
- Quality: The experiments suggest benefits to this approach, but the work lacks baselines from quality-diversity (like MAP-Elites) and relatively few evaluation domains for more rigorous comparison.
- Originality: Limited. Combining multi-objective optimization with RL is not novel: the quality-diversity literature (not cited) is focused on this topic.
- Reproducibility: Good. The appendix provides thorough details and the methodology is described with most details needed to reproduce the experiments. The experiment environments are provided, but not the code for algorithms.

**Strength And Weaknesses:**

## Strengths

1. Improves over widely used baselines. The study results demonstrate benefits relative to reasonable baselines in two different domains.
2. Straightforward to test and use. The algorithm is not complex or difficult to implement, making it a potentially strong addition to the set of multi-objective optimization approaches used in RL training.


## Weaknesses

1. Some missing experiment details. Were opponents weighted by algorithm type for RQ3? More details below.
2. Few domains. Pong is a relatively simple baseline domain with very limited degrees of freedom for agent actions. Justice Online is clearly complex and valuable. It would help to include results from one or two other benchmark competitive RL domains to better clarify the applicability and power of BiO over baselines. Right now it is hard to tell how much BiO is particularly tailored to Justice Online compared to being generally powerful.
3. Missing baseline: domain randomization. This is a minor point, but one of the main differences in performance relates to performing poorly in "different environments" (RQ2). The widely used practical solution is to randomize agent starting state during training. The results would benefit from reporting how all algorithms perform when applying randomization to the starting states during training. How much does BiO improve over this simple baseline augmentation?
4. No comparison to quality-diversity algorithms. The paper would benefit from comparing to a representative algorithm from the class of quality-diversity algorithms, as they are widely used for RL tasks (references below). MAP-Elites is one classic candidate.


## Feedback & Questions
- Q: For RQ3 the experiment picks a random opponent 60 times from a pool of 65 possible opponents (as I understand from the text). This seems likely to be highly imbalanced as 30 opponents are from EMOGI and 30 from BiO. Were the opponents selected so that 1/5 of the time an opponent comes from each of the algorithms? It would also help the results to include a matchup table reporting winrates of each algorithm against each variant, not only the average as reported in Table 3.

- Table 3: Why might PPO do so well in the Justice Online different environment (45.9% win rate) compared to the built-in AI (2.4% win rate in Table 2)? This may be a minor anomaly, but is surprising.

- Q: How would this approach scale to games where playstyle has high dimensionality? The experiments only examine a single dimension of style (distance to opponent). As dimensionality increases the algorithm would likely need to maintain a larger population and would sample each opponent less frequently. What do these scaling properties look like? Is there a test environment or setup that could examine this case?

- Q: Are there other arenas in Justice Online? The "different environments" setup is a change in initial position. How well does a baseline PPO algorithm do if trained with randomized opponent starting positions? Why does the starting position change cause such large differences in performance? This seems like the agents overfit to a very narrow range of conditions, rather than learning the "real" task, likely due to self-play having little incentive to mirror.

references
- MAP-Elites
	- Jean-Baptiste Mouret and Jeff Clune.  Illuminating search spaces by mapping elites. arXiv preprint arXiv:1504.04909, 2015.
- Other quality-diversity approaches
	- Pourchot, Aloïs, and Olivier Sigaud. "CEM-RL: Combining evolutionary and gradient-based methods for policy search." arXiv preprint arXiv:1810.01222 (2018).
	- Khadka, Shauharda, and Kagan Tumer. "Evolution-guided policy gradient in reinforcement learning." Advances in Neural Information Processing Systems 31 (2018).
	- Khadka, Shauharda, Somdeb Majumdar, Tarek Nassar, Zach Dwiel, Evren Tumer, Santiago Miret, Yinyin Liu, and Kagan Tumer. "Collaborative evolutionary reinforcement learning." In International conference on machine learning, pp. 3341-3350. PMLR, 2019.
	- Parker-Holder, Jack, Aldo Pacchiano, Krzysztof M. Choromanski, and Stephen J. Roberts. "Effective diversity in population based reinforcement learning." Advances in Neural Information Processing Systems 33 (2020): 18050-18062.
	- Jung, Whiyoung, Giseung Park, and Youngchul Sung. "Population-guided parallel policy search for reinforcement learning." International Conference on Learning Representations, 2020.
	- Fontaine, Matthew, and Stefanos Nikolaidis. "Differentiable quality diversity." Advances in Neural Information Processing Systems 34 (2021): 10040-10052.

**Summary Of The Paper:**

The paper proposes to optimize a population of agents generated through self-play to cover a multi-objective Pareto front among objectives for play style and performance (skill level). Play style is quantified in terms of state changes during play against various opponents. The optimization algorithm adapts the NSGA-II evolutionary algorithm to use a population generated through self-play. Empirical studies on two games shows benefits to the proposed algorithm when the starting state of a game is randomized and relatively strong performance against a pool of other agents (in both cases compared other RL or multi-objective optimization algorithms).

**Summary Of The Review:**


The technical novelty of the paper is the incremental change to NSGA-II adapted to self-play. Empirical results are strong on what is reported, but the base of comparison is narrow: two domains and without comparison to a related class of multi-objective optimization algorithms (quality-diversity). The style results show some diversity, but in a relatively narrow sense: navigating to a fixed point or circling around a space (with no results reported on Pong). I greatly appreciate the conceptual simplicity of the algorithm and see promise in the results. But the lack of a breadth of empirical results makes it hard to see the results as robust to a wide variety of competitive RL tasks. Ultimately the results are promising and with further validation would make a potentially useful contribution to the set of competitive RL algorithms.

---

### Official Review · Reviewer_1h9y · 2022-10-23

**Confidence:** 4
**Correctness:** 4
**Technical Novelty And Significance:** 3
**Empirical Novelty And Significance:** 3
**Recommendation:** 6

**Clarity, Quality, Novelty And Reproducibility:**

**Clarity**: good for the majority of the paper, with a few exceptions detailed below.

**Quality**: looks promising, but require some more details for experiments: see detailed comments below.

**Novelty/originality**: to the best of my knowledge it is sufficiently novel, and the obvious related work is discussed. I am personally not 100% familiar with all the literature on quality-diversity approaches and similar work, so I might have missed something here.

**Reproducibility**: Sufficient detail provided that I think reproducibility would be possible, though I saw no mention of source code.

---

**Detailed Comments**:
- Reducing "playing style" to a single scalar seems like a huge simplification, and potential limitation. I don't necessarily mind it, especially not when it turns out to work well, but it seems like something very obvious that should at least be acknowledge and discussed in multiple places in the paper: probably already in section 1 or 2, or at least definitely in section 3, and again also in the conclusion (where expanding on this to make  playing-style at least vector-valued could be a direction for future research).
- Below Equation (4), I find the description of $var_{\pi_k}(s_t, s_{t+1})$ unclear. What does "the relevant state change" mean? The "e.g., the position change of the agent $\pi$" part at least provides some intuition, but that only actually makes sense if we think of games where agents have a single "position" (e.g., it would make no sense in StarCraft). It is also not immediately obvious at this point in the paper why changes in position would be a good indicator of playing style, until further examples are given in later paragraphs. Personally, I would rather describe this thing as, for example, "a game-specific function quantifying how big the state transition was". I also don't really understand the notation with only $\pi_k$ (but not $\pi$) appearing in the subscript, since this is generally a function of both of them (but sometimes actually of only $\pi$!).
- The paper does not explain what "reward weights" are.
- For Tables 1 and 2: how many games were run to produce these win percentages? What are the (e.g., 95%) confidence intervals?
- I think the results in Figure 2 are very interesting, but have one potential concern, which is that only 3 different agents were picked. For all I know, these might have been cherry-picked to produce the most interesting plots. In Figure 2b, I can see that there are several other agents that are very close to the ones that were picked. For the Aggressive agent, there are also a few even more to the left (even more aggressive). While I understand it would not fit in the main paper, in the Supplementary Material it would be very useful to see similar plots repeated for a few more of those agents (especially one of the even more aggressive ones to the left), just to get an impression of how consistent these results are.

**Other minor comments**:
- p.7: what does "1G frames" mean?
- p.8: PPT --> should be PBT I guess?

**Strength And Weaknesses:**

**Strengths**:
- Majority of the paper well-written and easy to read.
- Interesting and simple approach, empirical results look promising.

**Weaknesses**:
- While I listed the majority of the paper being easy to read as a strength above, at the same time there is a weakness in that a few parts require more clarification. I provide detailed comments below.
- The experiments needs some more details provided, such that we can better quantify the significance of the results. See detailed comments below.

**Summary Of The Paper:**

This paper proposes a population-based self-play RL approach (with PPO for training in the inner loop, and an evolutionary approach in the outer loop) that trains a population of agents for games that are high-skill but also diverse in terms of playing style. First, it summarises "playing style" as a single scalar, which ranges between 0 ("defensive") and 1 ("aggressive"). Then, it aims to optimise a multi-objective (specifically, 2-objective) problem, where one objective adds the playing style to the skill level, and the other objective subtracts the playing style from the skill level. The approach is evaluated in 2 games (Pong and Justice Online).

**Summary Of The Review:**

A good paper in general, with a few important details that require some more clarification.

---

### Official Review · Reviewer_mGvT · 2022-10-31

**Confidence:** 4
**Correctness:** 2
**Technical Novelty And Significance:** 2
**Empirical Novelty And Significance:** 2
**Recommendation:** 3

**Clarity, Quality, Novelty And Reproducibility:**

The paper is clear and mostly well written. I would argue that adding PPO's equation doesn't  help readers who aren't familiar with the method and it doesn't teach anything new to readers already familiar with the method. The paper would be better off without Equation 1.

The appendix provides enough information to possibly reproduce the results from the paper.

**Strength And Weaknesses:**

Strength

This paper deals with a challenging topic, which is the one of learning strategies for playing complex games. The paper works with the underlying hypothesis that one is able to learn stronger strategies while keeping a population of diverse agents. While other population-based works dealt with similar research question, the paper does a good job reviewing some of these works. The experiments in Pong and Justice Online seem to support this underlying assumption.

Weaknesses

A key weakness of the paper is to not discuss the paper "A Unified Game-Theoretic Approach to Multiagent Reinforcement Learning" by Lanctot et al (2017). The paper talks vaguely about "self play" without explaining exactly which self-play algorithm they are referring to. Lanctot et al. describe a general framework for multiagent RL that allows one to instantiate algorithms such as Iterated-Best Response, Fictitious Play, and Double Oracle. As far as I know these are all self-play algorithms. In particular, Fictitious Play can be seem as a population-based algorithm and should probably be used as a baseline in the experiments. Why not consider these algorithms as baselines?

The paper is also constructed under the assumption that "maintaining an agent population does not necessarily mean maintaining diverse playing styles." (see Related Work section). However, population-based agents such as AlphaStar were designed to learn a diverse set of playing styles. It isn't clear from the experiments whether the proposed method is able to produce "more diverse agents" than algorithms such as AlphaStar.

Another weakness is related to how opponents are sampled to generate the results shown in Table 3. The experiments starts with a pool of agents formed by 1 PPO agent, 1 PPT (this is probably a typo and authors meant PBT) agent, 3 Multi-Reward Agents, 30 EMOGI agents, and 30 BiO agents. Then, an opponent is randomly sampled from this pool of agents for each of the evaluated methods. The agents play a game and the result is stored. The process is repeated 60 times and the average win rate is reported in Table 3.

This experimental design is problematic because it gives an advantage to EMOGI and BiO as they represent a much bigger share of the pool of agents. That way, both EMOGI and BiO are likely to face, during test, agents they were trained with. Despite this significant disadvantage, PBT is still competitive with BiO. This makes me wonder whether BiO is bringing something new in comparison to other PBT agents.

**Summary Of The Paper:**

The paper introduces a bi-objective optimization method for learning strategies for playing games from a pool of diverse agents. The method is population-based in the sense that it maintains a population of agents that attempts to optimize for a bi-objective function. Such a function accounts for the diversity of play, which is game-dependent, and winning rate.

The system was evaluated in the games of Pong and Justice Online, where the proposed method was competitive in some settings and stronger in others with respect to other population-based agents and RL baselines.

**Summary Of The Review:**

Paper deals with a challenging and important problem, but misses an important part of the literature that could be used as baseline in the experiments. The empirical design seems to be problematic as it might be giving advantage to the method introduced in this paper.

---

### Decision · Program_Chairs · 2023-01-20

**Decision:**

Reject

**Justification For Why Not Higher Score:**

Not seriously considering or comparing to important previous work.

**Justification For Why Not Lower Score:**

N/A

**Metareview: Summary, Strengths And Weaknesses:**

The idea of explicitly maintaining diverse styles in self-play is good, but not entirely novel. The reviewers list various strands of important previous work that are not considered in this paper. (Oddly, the authors mention Quality-Diversity, but seem not to have really understood it - "novel behaviours that we search for do not always come with high skill levels", but that is exactly the issue that quality-diversity is addressing.) Additionally, reducing "playing style" to a single value does the concept a large disservice; see, for example, the literature on procedural personas. The lack of serious consideration of related work, and comparing with such methods, impacts the paper negatively.